# Exploring the Mental, Social, and Lifestyle Effects of a Positive COVID-19 Infection on Syrian Refugees in Jordan: A Qualitative Study

**DOI:** 10.3390/ijerph191912588

**Published:** 2022-10-02

**Authors:** Khalid A. Kheirallah, Bayan F. Ababneh, Heba Bendak, Ahmed R. Alsuwaidi, Iffat Elbarazi

**Affiliations:** 1Department of Public Health, Medical School of Jordan, University of Science and Technology, Irbid 22110, Jordan; 2Clinical Psychology Department, Swinburne University of Technology, Melbourne 3122, Australia; 3Department of Pediatrics, College of Medicine & Health Science, United Arab Emirates University, Al Ain P.O. Box 15551, United Arab Emirates; 4Institute of Public Health, College of Medicine & Health Science, United Arab Emirates University, Al Ain P.O. Box 15551, United Arab Emirates

**Keywords:** refugee, COVID-19, camps, community, healthcare, well-being

## Abstract

Migrants and refugees are among the vulnerable populations that suffered disproportionately from the COVID-19 crisis. However, their experiences with COVID-19 positivity status have not been investigated. This study explored the physical, mental, and psychosocial impacts of a positive COVID-19 diagnosis on Syrian refugees living in Jordan. Using a qualitative approach, twenty phone interviews were conducted with ten adult Syrian refugees living within the camp and ten refugees living in non-camp (host community) settings in Jordan. Follow-up interviews with five health care providers at a refugee camp were conducted to explore the services and support provided to the refugees with COVID-19 infection. The findings were thematically analyzed and grouped into major themes, subthemes, and emerging themes. Refugees living within camp settings had better access to testing, healthcare, and disease management and did not experience fear of being deported. Refugees in both settings suffered mental and psychosocial health impacts, social isolation, fear of death, and disease complications. COVID-19 infection has negatively impacted refugees’ well-being with noticeable disparities across the different living conditions. Refugees living within host community settings may need more support for managing their condition, accessibility to free testing, as well as treatment and healthcare services.

## 1. Introduction

Worldwide, about 600 million people have been infected with severe acute respiratory syndrome coronavirus-2 (SARS-CoV-2), the causative agent of the Coronavirus Disease 2019 (COVID-19) as of August 2022 [1]. Initially centered in the Wuhan Province of China, the virus has spread globally and resulted in close to six million deaths across 215 countries [2]. The clinical implications of the virus are variable and range from mild, flu-like symptoms to severe respiratory distress and organ failure disease [3]. Expectedly, health professionals have focused on the medical impacts of the virus while governments have focused on Non-Pharmaceutical Intervention (NPI) measures and vaccine administration when available. The NPIs are public health measures that aim at infection containment using physical distancing and restrictions on movement, travel, school, work, and other non-essential activities [4]. While this is crucial for controlling the spread of the virus, research from previous pandemics has found that restrictive measures and perceived risk of infection resulted in negative impacts on psychological and social health such as depression, anxiety, posttraumatic stress disorder [5,6,7], loneliness, isolation, anti-social behaviors, domestic violence, and child abuse [8,9,10].

Preliminary research has found that the psychological and social health of vulnerable populations has been severely impacted due to COVID-19-related NPI measures [11,12,13,14]. Yet, research studies that explore the specific challenges associated with a positive COVID-19 diagnosis in vulnerable populations are limited. Refugees are a particularly vulnerable group during the current pandemic due to their living conditions that increase their risk of being infected [15]. For example, preventive health measures recommended to control the transmission of COVID-19, such as hand hygiene and physical distancing, are not possible to implement and/or sustain in highly dense and poorly maintained areas, such as refugee camp settings [16]. Further, the majority of refugees currently reside under temporary living conditions within poor urban settings that suffer crowdedness, shortage of clean water, and personal hygiene equipment [16]. Shortage of public health services, financial means to support their families, and lack of proper sewage systems are additional challenges for the spread of COVID-19 among refugees. This increases their susceptibility to communicable and non-communicable diseases [17]. Additionally, it may be more difficult for refugees to follow public health advice, as quarantining or self-isolating the COVID-19 positive cases can exacerbate the monetary challenges that refugees face on a daily basis. Such a socio-economic burden may further contribute to spreading the infection within refugees’ households and local communities.

In Jordan, a country that hosts around 670,000 United Nations High Commissioner for Refugees (UNHCR) registered and 500,000 unregistered Syrian refugees, the first cases of confirmed COVID-19 were reported in Zaatari and Azraq Refugee Camps in September 2020 [18,19,20,21]. While 80% of Syrian refugees live within the host communities outside of refugee camp settings, it is currently not known how many refugees outside of the camps have tested positive for COVID-19 [18,21]. The most recent report by the UNHCR revealed that, as of February 2022, approximately 116,000 cases of COVID-19 were reported among forcibly displaced people, and up to February 2022 [19], around 2400 refugees tested positive for COVID-19 inside refugee camps in Jordan [18]. During this period, the figure was below the national average, as 2% of refugees tested positive, compared to 4% of the total population. However, this figure may have increased as Jordan had a spike in cases in late March 2021, reaching the highest recorded number of cases per day [1]. According to UNHCR reports, over 90% of the Syrian refugee population in camps in Jordan and about 50% of the Syrian refugee population in urban areas (18 years and above) have received at least one dose of the COVID-19 vaccine [20].

Syrian refugees living in Jordan, within the camp or host community settings, suffer from economic disadvantages and poor living conditions as described previously [21]. A vast majority of these refugees live outside the camps in different cities and areas, most of whom have no work permits, forcing them to rely on humanitarian assistance to survive [22]. It is estimated that 86% of those living outside camps live in poverty [22]. A COVID-19 infection and its related NPI measures seemed to have complicated the situation and added fragilities to their living conditions [21]. There are initial indications that refugees may refrain from seeking medical support due to fear of the stigma associated with a positive COVID-19 diagnosis, including the guilt of contracting the virus and spreading it to others [21,22,23,24,25]. Aid workers in Rohingya camps in Bangladesh reported minimal testing amongst refugees with COVID-19 symptoms, as they feared isolation from their families, deportation, and even being killed [26]. Consequently, the social stigma associated with COVID-19, in general, may result in symptom concealment, a delay in receiving early treatment, and a lower likelihood of compliance with public health measures [27]. This may also result in clinically significant mental and social health effects [24,25]. It is thus crucial to explore and assess the COVID-19-related social and psychological challenges specifically faced by refugees. This will help inform public health interventions and policies and curb the spread of the virus. In this study, we aim to specifically explore the physical, mental, and psychosocial impacts that a positive COVID-19 diagnosis has on Syrian refugees living in Jordan, within the camp and host community settings, utilizing a qualitative study design.

## 2. Methods

This study is reported according to the COnsolidated criteria for REporting Qualitative research (COREQ) [28].

### 2.1. Theoretical Framework

An interview guide was created based on (Lazzarus and Folkman 1984) the coping strategy model [29] and (Burr and Klein 1994) the conceptual framework of coping strategies [30]. The principal investigator (IE) developed the interview guide (Appendix A). The other research team members reviewed the interview guide and agreed on the major questions and prompts. Ethical approval was obtained from the Institutional Review Board of Jordan University of Science and Technology (24/139/2021).

### 2.2. Sampling

A purposive and convenience sample of 20 refugees with recovered COVID-19 infection, and five health care providers (HCP) from the Zaatari camp, were included in the study. Snowball sampling technique identified refugees who had a COVID-19 infection between February 2020 and March 2021.

### 2.3. Recruitment

Participants were first identified through workers within healthcare facility services provided to refugees within both settings. Once potential participants were located, they were contacted by phone. All interviews were audio-recorded and then transcribed into verbatim text.

The Researcher who conducted the interviews (BFA) had access to the refugee camp through a physician who worked there. The physician invited refugees to participate in the study through his encounter with them. Those who agreed were referred to the researcher who contacted them, obtained their consent to participate, and then interviewed them. As for those in the community (non-camp), they were recruited through the snowball technique. Upon initial approval, participants received the study information package (via WhatsApp) to review and provide an initial agreement to participate. Those who agreed to participate were contacted to schedule a phone interview. A qualitative approach was utilized, and in-depth phone semi-structured interviews were used. Around 30 participants were invited, but only 20 were accepted to participate in the interview. Reasons for refusal included lack of time or simply being not interested. A sample of 20 Syrian refugees, ten males, and ten females, were interviewed over the telephone due to COVID-19 restrictions. Ten of the refugees lived within refugee camp settings, and the other ten lived within host community settings. An interview guide was created to explore the potential impact under investigation. Only adult participants who self-reported being diagnosed with and recovered from COVID-19 infection between February 2020 and March 2021 were interviewed.

Additionally, we interviewed five healthcare professionals (HCP) who were directly involved with the healthcare of refugees within camps to explore the management, services, and support for positive COVID-19 cases. The interviews were conducted in April 2021. Five HCPs (3 Medical doctors and 2 nurses from the Zaatari camp) were interviewed over the phone by the same investigator (BFA) who conducted the interviews with the participants. Participants were approached through the snowball technique. The HCP participants were approached by the same worker who helped in identifying refugees. We planned from the beginning to interview five HCPs to explore the services and support provided for positive COVID-19 patients in the camp. The study information package shared with participants included information about study aims, voluntary participation, confidentiality, the nature of questions to be asked, and a statement that personal information will not be collected. Each interview lasted between 40 and 60 min.

### 2.4. Reflexivity

IE, an experienced qualitative researcher, and the study PI trained a research assistant (BFA) via zoom to conduct interviews and thematic analysis and identify themes. IE holds a DrPH in qualitative research methods. IE supervised the entire process of the study. To ensure researchers’ reflexivity, some of the interviews’ transcripts were returned to participants by WhatsApp for review and approval. Participants showed no concerns and were satisfied with the transcribed words. The PI and the research assistant reviewed and verified the transcripts to ensure transparency and minimize bias.

### 2.5. Data Collection

#### 2.5.1. The Interview Guide

The interview guide was developed in Arabic and English with questions exploring the mental, social, psychological, spiritual, and lifestyle impacts of a positive COVID-19 diagnosis on participants. The research team reviewed the interview guide and agreed on the final questions and concepts to be addressed. Interviews were conducted in Arabic by trained researchers. A research assistant collected all recordings, transcribed, and translated them from Arabic to English. All interviews were imported to NVivo 12 [31] for analysis and coding. Data saturation was reached after interviewing the ten participants from each group.

#### 2.5.2. Health Care Professionals’ (HCP) Interviews

The HCPs’ interviews included open-ended questions that covered mainly: management protocols for COVID-19 cases, classification of cases, available vaccines, availability of vaccines, Polymerase Chain Reaction (PCR) testing, dealing with suspected cases, availability of medical services, coverage by insurance, refugees’ behavior toward following safety precautions and contacting confirmed cases.

### 2.6. Data Analysis

A thematic analysis using Braun and Clarke’s six step process was performed [32]. A thematic tree was created to highlight the most common themes (Figure 1). IE and BFA extracted themes individually, agreed on the final themes, and then verified them with other research team members.

The interview guide was divided into questions related to physical, social, mental, lifestyle, and spiritual effects of a COVID-19 positive diagnosis on the participant. The coding was based on these five major nodes.

#### Derivation of Themes

Theme derivation was based on the reported effects as described by participants. Node one: Physical impact included symptoms experienced by participants. Node two: Social impact consists of the implications of COVID-19 diagnosis on social relations with direct family, members, and friends during the infection period. Node three: Mental health impact consists of the impact on psychological status during and post COVID-19 diagnosis and returning to everyday life. Node four: Lifestyle impact consists of the impact on dietary supplement use, diet modifications, sleep and exercise, and other lifestyle factors. Node five: Spiritual impact includes the relationship with God, beliefs and faith, and views toward death and diseases.

## 3. Results

The analysis included ten refugees living in one camp (Zaatari camp, Jordan) and ten living within host community settings. Participants’ ages ranged from 20 to 56 years and 50% were females (Appendix A). Themes and subthemes were organized based on the main aims of the study. For each major node, a set of themes are listed (Table 1). Some emerging themes were identified and are presented by setting (i.e., in the camp and in the host community) and listed in Table 2. Finally, we present in Table 3 the frequency of themes among participants in both settings. The following section will present the major themes, emerging themes, and the health providers views.

### 3.1. Major Themes

#### 3.1.1. Node One (Physical Health)-Themes

##### i. Theme One: Signs and Symptoms

The physical impact of a positive COVID-19 diagnosis on the refugees, within camp and host community settings, was not different. None of them required hospitalization. The usual symptoms of COVID-19 reported were mainly: loss of taste and smell, malaise, fatigue, fever, flu-like symptoms, and severe headache. Few indicated that the loss of taste or smell did not last long while some stated that their concerns were not regaining their original sense of smell and having an unpleasant smell most of the time. Most of the participants in the camp had mild to moderate COVID-19 disease.

For example, **Com01F** said:” I did not have many symptoms, just pain in my joints and loss of my taste and smell sensations”.

While **Camp10F** said: “I developed fever for three days; my children and husband did not develop any symptoms”.

The physical impact at the time of the interviews was discussed being as short term and most of the participants reported only the short-term effects that were later recorded in the literature as the most common COVID-19 symptoms.

##### ii. Theme Two: Medical Support

Refugees living within camp settings reported good medical support provided to them by default, being in the camp with accessible clinics and medical teams, with open access for testing and pharmaceutical support. However, those in host communities reported not seeking medical assistance due to financial reasons, although they had symptoms. For example, one participant did not perform the COVID-19 test despite having symptoms and knowing that it was COVID-19 as her family members had a confirmed COVID-19 diagnosis before.

**Com05F** said:


*“I had symptoms like my other family members but preferred not to get tested to save money. “I did not need to go to the hospital or receive medical treatment”.*


Those in the camp had access to medical services, including medications, which reduced their anxiety and the way they were dealing with the experienced symptoms. This was reported by most of the camp refugees. For example, **Camp07F** said:


*“I was not concerned much because I knew that I would be looked after by the health care team and by my neighbors.”*


#### 3.1.2. Node Two (Psychological Health)-Themes

##### i.a. Theme One: Stigma and Blame

The mental and psychological impact of the COVID-19 positive status was similar between both groups, in terms of the stigma and fear of being identified as a positive case, while most expressed their anger, anxiety, and fear of complications especially those who had COVID-19 during the earlier period of the pandemic.


**Camp09F:**



*“Here at camp, people are closedminded and consider who got the virus being an unclean person with poor hygiene”.*


Another from the community group, **Com03M** indicated that:


*“I did not want people to know that my daughter and I had COVID. I preferred to keep away from the neighbors so they do not know about our positive infection”.*


Moreover, people affected by this stigma tended to blame others such as **Com01M who** blamed his father for giving him and his sister the virus as he was not compliant with recommendations. He said:


*“I think we got it from our father since he was not compliant with safety precautions”. The participant indicated that the father passed away due to COVID-19 complications.*


##### i.b. Themes Two: Fear of Loss (Loved One or a Job), Fear of Complications

Many participants in both the camps and the host community expressed fear of losing their job and access to work, being casual workers who depend on their daily income. Moreover, some expressed fear and anxiety originating from losing their loved ones due to COVID-19.

A participant from the camp, **Camp08F** said:


*“I am annoyed because I lost my job from the day I got infected. My elder brother is giving me pocket money. I hope to find a new job soon”.*


Another participant from the community **Com04M** said:


*“I was upset and sad because I lost my salary during the isolation period and my wife borrowed money from her cousin to cover some daily needs”.*


Similarly, two participants expressed their frustration:

**Com03F** said:


*“At the beginning, I was annoyed because I lost my salary since I need money to help my family in daily life needs and university fees and to save some to get married, but thanks God I returned to my job”*


**Com05F** said: 


*“I was so sad that getting the virus forced me to borrow money from neighbors to buy food”*


For others, worrying about losing their loved ones was more frustrating. For example, **Com01M** said:


*“I was worried about my father and unfortunately we lost him.”*


Another participant; **Com04M** said:


*“I was worried only about my parents. They lost their smell and taste sensations. When I get my salary this month inshallah, I will refer them to the medical advice”.*



*This is considered to be a major concern for most of the participants the issue of losing loved one, the economic impact.*


#### 3.1.3. Node Three (Social health)-Themes

##### i.a. Theme One-Spouse Support

The social impact among refugees within camps varied. Some specified not receiving much support from their spouse, especially in household chores. One lady said that she was even mistreated by her husband. **Camp06F** said:


*“My husband was not supportive, and he got mad with me. He had to take care of the home and kids …. He did not really look after them properly…. Actually, I didn’t receive good family support like any other infected persons and that made me sad ….and I was crying all the time”.*


The spousal support was reported by the camp residents while it was not an issue among those in the host community.

##### i.b. Theme Two. Seeking Support

Many refugees within camp and host community settings said that they resorted to calling family and friends in Jordan or Syria to access support and to spend their isolation time. Some used spiritual practices, reading, studying, internet and television to cope with the situation. Some expressed their over-protection of children, and husbands by isolating themselves and by providing them with herbal and homemade remedies to avoid complications. Some indicated that the COVID-19 period helped them to reconnect with family members and strengthened their bonds.

A lady from the camp said, **Camp06F:**


*“I had more time during isolation to talk to my sisters in Syria over phone, this was an advantage of being isolated”.*


Another **Com02F** said:


*“In the first 2 days of my isolation, I spent time on Facebook and WhatsApp. Then when my parents and brothers got the infection, I was cooking and cleaning the house”*


**Com03M** said:


*“The father of my daughter’s friend was calling us all the time and provided all support and advised us to use vitamins C and D”.*


**Camp08F** said:


*“I was busy doing my homework and studying”*


**Camp10F** said:


*“I was talking with my friends and sisters on phone, and spending time on YouTube listening to Quran”.*


#### 3.1.4. Node Four-Spirituality Themes

##### i. Theme One: Closer to God

Most participants considered COVID-19 to have a positive impact on their spirituality. Many expressed their views that the experience made them closer to God and that they spent most of their isolation time praying, reading the Quran, and self-reflecting on their lives.

**Com01F** said:


*“We are Muslims and do our prayers all the time, but when I got the result, I prayed to God for protecting my family and me more”.*


**Com02F** said:


*“I had more time to listen to Quran and thanks to God, I am Muslim and do my prayers on time, and I have a good relationship with God”.*


A participant from the camp said, **Camp 08F:**


*“I spent my isolation reading Quran”*


**Camp06M** said:


*“We are Muslims, and we are close to God, but after getting the virus, I felt that I had to do prayers on time and to get closer to God...”.*


Moreover, spirituality was a coping mechanism for many.


**Com01M said:**



*“The first few days after my father death, I was sad but it’s a destiny by God and he was not compliant with safety precautions, he was 70 years old and he had hypertension and diabetes mellitus, I spent my isolation time reading Quran”.*


However, many of the participants stated that their spiritual practices and relation with God was always strong and that the COVID-19 did not impact on their belief and faith. As participants Com 03 M and Com 05 M and others said:


*“We are Muslims, and we are close to God”.*


##### ii. Theme 2. Acceptance

Many of the participants repeated the word thank God, even those who lost someone due to COVID-19. Accepting the infection as being from God and being “their destiny” was expressed by all participants.

**Com01M** said:


*“I had no feeling of panic or thinking about complications or death, the only thing I was worried about my father, and he developed respiratory problems and passed away; Alhamdullillah (Thank God)”.*



**Com02F said:**



*“I am Muslim and believe that this virus is from God like any other diseases and Alhamdullillah (Thank God) we recovered and feel good now”.*


#### 3.1.5. Node Five: Lifestyle

##### i. Theme One: Precautions and Protection

Most participants acknowledged the importance of preventive measures, a healthy immune system, and homemade remedies. Social support, helping each other, and checking on others were one of the most recurrent words expressed by participants. The COVID-19 situation created a great fear of loss of loved ones amongst participants. This was also expressed as a motive to change lifestyle and to protect oneself and loved ones from the infection.

Participants **Com05M, Camp07F, and Camp07M** similarly said:


*“We are reminding each other about following the safety precautions strictly; we are afraid of each other of getting the virus again. May God protect us and all people around the world.”*


**Com01M** indicated that he did not change his lifestyle except for increasing precautionary measures.

Additionally, **Com03M said** “My wife and I were calling each other all the time checking our health status, and thank God my family didn’t get the virus.”

**Com03M** said:


*“We all supported each other and I spent more time with my family since I was on a sick leave, we enjoyed our time together, Alhamdullillah (Thank God)”*


When asked about medical and pharmaceutical support during the infection, many participants indicated resorting to herbal treatments. Others expressed being unhappy that authorities did not contact or call them.

**Com03M** said:


*“We used to drink herbs like ginger and eat fruits and vegetables like orange”.*


**Com01F** said:


*“I received just paracetamol for fever and pain and used herbs like ginger and I was drinking orange or lemon as source of vitamin C”.*


**Com02F** said:


*“We received vitamins C, D, and zinc as my daughter friends used them since her father is a doctor at the ministry of health and prescribed that for her”. He also kept on checking on us …”*


The participant said: 


*“Yes, after getting the virus we changed our diet to be more healthy by increasing vegetables and fruits portions we kept our homemade food as before I got the virus, we don’t do exercises, I have same sleep hours I used to sleep more first few days of getting the virus then my sleep is normal like before getting the virus, about religiosity Alhamdulillah we are Muslims and do our prayers all the time but first two days of the virus I were used to pray to God for protecting me and my family”*


**Com03M** also said:


*“Nobody called us from the Ministry of Health, we didn’t need any medical care support but I was referred to an ENT physician due to losing my sensations, he told me if my sensations don’t return back within one year, I will lose it permanently”.*


Although more than half of participants inside and outside the camp believed that COVID-19 was an actual situation and not a man-made conspiracy, some thought the virus was man-made.

**Com02M** said: 


*“This virus was made at the laboratory to get rid of elderly people in the world, and don’t ask me why because this is my inner feeling.”*


**Camp08F** said:


*“It’s real and transmitted all over the world”.*


So many of the participants indicated that they used to live a healthy lifestyle and that they did not change their lifestyle much, while many said that the infection was a motivation to remember the precautions and for living a healthy lifestyle (eat healthy, sleep, and exercise).

**Com03F** said that:


*“After getting the virus and till this moment we are taking care of each other and reminding each other by safety precautions all the time”.*



*“We usually follow a healthy lifestyle even before COVID-19; nothing changed.”*


**Camp09F** said:

“… *my husband got mad because he lost his salary during the isolation period he needs his salary to buy cigarettes for smoking but thanks God after finishing the isolation he got relieved*”……She continued “…*I tried to calm him down but he was so mad because he was not able to buy cigarettes but Alhamdullillah after discharge he returned back to his work, but that didn’t affect me at all I was happy deep inside because he stopped smoking in isolation period*”.

**Com01F** expressed her concerns with the possible long-term effects of COVID; however, as her financial situation was not very good, she indicated that she needed to save to go to a doctor. She said:


*“Yes, I am concerned and if it continues, I will save money to visit a private doctor because I don’t have medical insurance”.*


##### ii. Vaccine Hesitancy

Many of the participants expressed their lack of trust in the vaccine trials and their effectiveness. It is important to remember that the study was conducted at the earlier stages of the pandemic. Vaccines were still being rolled out at smaller scales with lots of rumors and misinformation circulating. Additionally, efforts from authorities to fight the vaccine and the COVID-19 infodemic were still timid and not taken seriously. Many of the participants in both settings vocalized their concerns with the vaccines.

**Com03M** said:


*“…No, I think they are new and needs more studies to confirm their safety and efficacy”*


**Com01F** and **Com02F** said too:


*“No, I am still not convinced by the safety and efficacy of those vaccines maybe after years when they are safe and effective 100% and are free.”*


While **Camp06M** and **Camp010F** said:


*“I will take the vaccine however, they said it will take months until we receive it”.*


Similarly, **Com03F** indicated that she wants to take the vaccine, but she was told she needs to wait for months.

### 3.2. Emerging Themes

Emerging themes were classified based on setting (camp vs. host community) as presented in Table 2. Overall, while some subthemes seem to be similar by setting (such as Signs and Symptoms), the results suggest distinct sub-thematic differences between the two settings as indicated by the frequency of subthemes presented in Table 3. Access to good medical support, for example, was reported by all subjects within camp settings (n = 10) and only by three subjects in the host community. Similarly, lack of spousal support was reported by three participants within the camp setting, yet it was not reported by any participant within the host community.

Those who were living within community settings experienced financial and economic impacts from COVID-19. Many of them specified that such financial impacts affected their access to testing, medical help, and coping ability. They experienced anger and fear of losing their job and the possibility of losing their income. Mothers or females in this group had a strong feeling of responsibility to look after their spouses, the breadwinners, and their children.

Those within camp settings experienced more social exclusion, social issues, with husbands and neighbors, affecting their mental health and social relationships. Those within camp settings did not report problems accessing medical care and medications. They had no fear of being jobless; only one or two mentioned this issue and some mentioned being affected financially.

**Camp07M** said:


*“I received free treatment and received free coupons with 20 Jordanian Dinars for each one of us from UNCHR in the camp.”*


**Camp010F** said:


*“Here at camp, the environment is not healthy and closed-minded also, and they are not educated; they considered who got the virus to be a dirty person or with scabies.”*


**Camp06M** said: 


*“I am not upset due to not getting paid; as I told you, we are receiving help from UNCHR and live for free at camp, and our water and electricity are free.”*


**Com01M** said:


*“I lost my job from the date I got infected by the virus, my big brother is giving me pocket money, I want to search for a new job” …I wish I can find a job soon”*


Those outside the camp discussed the impact of the virus on their financial situation too.

**Com04F** said:


*“I preferred to isolate myself at home using home remedies to save costs for buying bread or chicken for her daughters”.*


**Com01F** said that her husband had to take on more shifts as a security guard to compensate for the loss of the money during their isolation.

**Com05M** said:


*“I was upset and sad because I lost my salary during the isolation period and my wife borrowed money from her cousin to cover some daily needs, but Alhamdulillah (Thank God) I recovered and returned back to my job”*


To provide a clear idea on the distribution of the themes per participants we are providing frequencies and percentages in Table 3).

### 3.3. Health Care Providers Interview Findings

Health care providers reported following up-to-date COVID-19 treatment protocols that were in line with the World Health Organization (WHO) and Jordanian Ministry of Health (MOH) protocols and the pandemic status in Jordan. They all claimed that refugees in the Zaatari camp received the best and free treatment for COVID-19 and even other diseases. They reported that, at the beginning of the pandemic, COVID-19 cases were quarantined in the Dead Sea before an isolation area was prepared within the camp site. They also reported that PCR tests are free and available all times for suspected cases; symptomatic or asymptomatic. Once a case is confirmed, it is classified according to severity (mild, moderate, or severe). Mild and moderate cases are treated in the camp while severe cases are referred to MOH hospitals outside the camp. Treatment, which mainly depends on the severity, include administering antipyretics, antibiotics, oxygen masks, and nebulizers. Interviewed HCPs also reported that the vaccination campaigns were initiated as that within Jordan using the same vaccines for Jordanians (Sinopharm, Pfizer, and newly added AstraZeneca), and that all medical services for COVID-19 cases and vaccinations in the Zaatari camp were provided free of charge. They all agreed upon the easier tracking system in the camp for close contacts of confirmed cases. However, they also agreed that some of the refugees hid their symptoms to avoid testing, isolation, and losing their jobs and some hid them to avoid embarrassment and stigma. Even some of the infected cases chose to be isolated at home because they have children with disabilities; hence, the health care team in the camp provided in-home visits and services.

## 4. Discussion

The current study explored the physical, mental, and psychosocial impacts of a positive COVID-19 case on Syrian refugees living within the camp and host community settings in Jordan utilizing a qualitative approach. The results indicated significant economic, social, and mental health impacts expressed by participants, which are associated with a positive COVID-19 infection. The impacts presented in this study do not appear to be uniform among refugees and to differ by settings (camp vs. host community).

The living conditions of Syrian refugees in Jordan are not optimal as the country does not support re-settlement programs [20]. Uncertainties associated with Syrian refugees are expected to have been further complicated by the spread of COVID-19 and NPI measures [33]. This has added to the disease’s economic, social, and mental health burden among such vulnerable groups [20,21,22]. It is necessary to investigate the overlapping fragilities associated with COVID-19 infection among Syrian refugees to reduce the associated burden impacts. Public health interventions should consider such differential burdens and focus on reducing the social stigma associated with COVID-19 and the economic barriers associated with it.

The impacts of COVID-19 on Syrian refugees living in Jordan seem to overlap and interconnect and may not be addressed separately as identified in this study. Physical effects, for example, intersected with social stigma and economic burden, which may have negatively affected social support as reported by participants. A COVID-19 infection for a refugee indicates the loss of a temporary, low-paying, job and a significant reduction in ones earning potential and ability to meet essential needs [33,34]. This significant stressor adds to the fragility of living conditions of refugees and exposes them to a burden that was not expected [33,34,35]. This is mainly for refugees within host communities, as they still have to pay rent and provide daily living essentials. In contrast, camp refugees have access to more humanitarian aid and did not have to pay rent. Moreover, refugees in the camps had better access to health care services, as reported by participants. This is supported by El-Khatib et al. (2020), who argued that the Zaatari camp adapted many supportive mental health activities to support refugees during the COVID-19 pandemic [36].

Availability of healthcare services within camp settings seems to buffer the physical impact by eliminating the financial needs required to secure testing and medical expenses. This is even though humanitarian funding has been reduced due to the global effects of COVID-19. Still, Jordan was one of the countries that included refugees in its national response to COVID-19, including testing, care, and immunization [36,37]. In our findings, refugees in the camp seemed to exhibit more vaccine hesitancy than those in the community. Studies have reported vaccine hesitancy among those who are underprivileged and of lower socioeconomic status and education in general and among refugees too [38,39,40,41]. It is known that refugees in camps may fit within this social description, and this could be the reason why it was reported to be more common among our subjects. In contrast, the immunization program coverage is high among refugees in camps in Jordan, while refugees in urban areas received less coverage. This should also be addressed through more education and better accessibility. Overall, the social structure seems to be a critical dimension when addressing refugees within camp and host community settings. This was more evident when spousal support was not reported as part of the subthemes within host community while it was reported by three participants within the camp (Table 3). Such differences call for further investigation of the social dimension to be able to better identify the difference in the social structure between refugees living within camp and within host community settings. More research is also needed to understand factors that may affect vaccine hesitancy among refugees in camps and those who live in host communities. Moreover, in our study refugees in camps reported poorer spouse and social support which was reported in earlier studies [42].

In light of our findings, refugees outside the camps may suffer more from the physical impact of COVID-19 due to their poor financial and socio-economical statuses. It was reported that access to health care services is good to those refugees registered in Jordan this may be not the case for those living in the host community who might not be registered [43]. While there is a global call for more attention to the mental and psychological impact of COVID-19, refugees seem to be more vulnerable to such impact given their prior exposure to conflict-related trauma [37]. Anxiety, anger, and fear among refugees are well-established given the traumatic exposure they witnessed in Syria, during the refugee journey, and while living as a refugee [37]. This adds to the burden of mental health problems among refugees and puts them at higher risk of long-term impact [37,44]. Of note, as well, is that this impact seems to be universal regardless of the settings (camp vs. host community). Surveillance, screening, and fine-tuned interventions for the effects of COVID-19 may be a critical need to avoid the potential adverse impacts of complicated, intersected stressors on mental health wellbeing [45].

The social impact seems to be rooted within the Middle Eastern culture, as the family is the center of attention from caregivers [46]. Families in such cultures provide social support to those in need and ensure attention is provided when needed. Alzoubi found in pre-COVID-19 era that Syrian refugees had different coping strategies with other predictors [43]. The authors found that seeking social support was associated with “being female, older, and widowed; having a lower education and lower total income; being dissatisfied with their income; being non-employed and having chronic illnesses”. Gender roles also seemed to emerge during COVID-19 as wives did not perceive social support from their husbands during the COVID-19 infection period. Regardless, the social dimension of COVID-19 infection may have presented times for refugees to cope and provide social support when needed. At the same time, isolation and exclusion relate more to the perceived stigma associated with COVID-19 infection and have been the case among refugees and at the global level [27]. In our study, the health care providers also reported that stigma played a role in people hiding their symptoms and not seeking treatment as reported by few of our participants in both settings. However, it was a more reported theme among those in the host community due to the fear of being identified and for economic reasons The United Nations Women Jordan Rapid Assessment Report demonstrated that tensions within households were reported by refugees in camps as well as economic hardships [43]. Bianco and Cobo (2019), pointed out to the impact of being Syrian refugees in Jordan on education attainment and literacy due to the poor access, impact of exposure to trauma and violence and other social factors such as the way teachers treat these [47]. This may not have been investigated by this study; however, the COVID-19 situation may have worsened this crucial human right for refugees due to isolation and the stigma of being infected [48]..

Social stigmatization of COVID-19-infected individuals seems to have potentially flourished with dramatic stories in media and on the internet [49]. Worldwide, several features of Stigma toward individuals of Asian descent, those with recent travel history, and healthcare professionals have also been reported [49]. Migrants and refugees were identified to suffer from many consequences during crisis [50,51,52]. It was reported that refugees in Europe suffered from discrimination [53] but to our knowledge, no study reported on COVID-19 related stigma among Syrian refugees in Jordan Social Stigma associated with COVID-19 infection may be attributed to the disease being new and not fully perceived among the public [52]. In Jordan, the first cases of COVID-19 were severely stigmatized through being labelled, stereotyped, discriminated against, and treated separately [53]. This stigmatization was augmented by the lack of accurate information and the widespread misinformation about the disease [53]. This infodemic became a global phenomenon that may have complicated the social acceptance of the disease and presented it as a reason for stereotyping and discrimination against those infected or those who made contact with infected individuals [54]. Accordingly, it is expected that Syrian refugees being stigmatized due to COVID-19 infection seek immediate healthcare services less frequently and be discouraged from adopting health behaviors to prevent the spread of the disease. This calls for more empathy to be shown to COVID-19 patients and ensuring that the public understands the disease itself and facilitating the adoption of measures to help infected cases keep themselves and their loved ones safe and infection-free. While this may not be feasible given the living conditions of refugees within camps or host communities, self-isolation and community support for quarantines could be a solution using social norms and local health experts from within the refugee population. The public should also be informed about the potential role of stigmatization on individuals and the community and the need to have them play a role in preventing discrimination through kindness, speaking up against negative stereotypes, learning more about mental health, and sharing individual experiences to provide the support needed. Caregivers and local leaders working with refugees should also encourage dispelling fears, misconceptions, and the stigma associated with COVID-19. As necessary, healthcare providers should consider social stigma among refugees as a significant mental health problem that could be exacerbated by the socioeconomic hardship caused by the pandemic. This will complicate refugees’ integration and increase their living uncertainties.

## 5. Limitation and Strengths

This is the first study to explore the impact of a positive COVID-19 infection on the wellbeing of Syrian refugees within camp and host community settings. It is the first to explore views, beliefs, and positive or negative impacts of such infection in a holistic approach. Limitations include using the phone for the interviews instead of face-to-face, which was due to the COVID-19 mitigation measures. In fact, this is why the sample interviewed was low. The time limit and the difficulty to reach refugees in both settings were major contributors to our low sample size. This is especially true given the COVID-19 limitations on social encounters and face to face research. As in qualitative studies, generalization beyond the study participants and small sample size are still an issue. More quantitative and qualitative studies on a larger scale would provide a better picture with less risk of bias. A longitudinal study approach may also facilitate assessing the impact of COVID-19 on the long run and fine-tune interventions to avoid the potential life-long implications of the possible effects on the refugees. Still, this study presented, for the first time, the potential impact of COVID-19 infection among vulnerable groups that are not commonly investigated. This is a great potential to pay more attention to refugees during public health emergencies and to further our understanding of the potential impacts of such emergencies on their life. Syrian refugees’ health is understudied in general, more exploratory studies are needed to understand this population physical, psychosocial, environmental, and economic needs during crisis and beyond that.

## 6. Conclusions

A COVID-19 positive infection had various influences and effects on Syrian refugees in Jordan. Being in the camp may be protective for the social and physical health of refugees while living outside the camp may force refugees into more isolation and may impact their physical, mental, social, and economic health. Spirituality, however, was suggested as a protective factor for both groups. Living within the camp or host community settings can be considered an essential determinant of how a positive COVID-19 diagnosis may impact the health and wellness of a Syrian refugee. Support for those who test positive from local authorities and UNCHR is highly needed to reduce the burden on this population and their families. Studying further the conditions of both groups and how COVID-19 infection may have impacted on this vulnerable group or on how further it will impact is highly recommended. The growing number and the emergence of more variants may have worse effects on refugees especially with more efforts and resources allocated to ward the general community rather than specific vulnerable groups.

## Figures and Tables

**Figure 1 ijerph-19-12588-f001:**
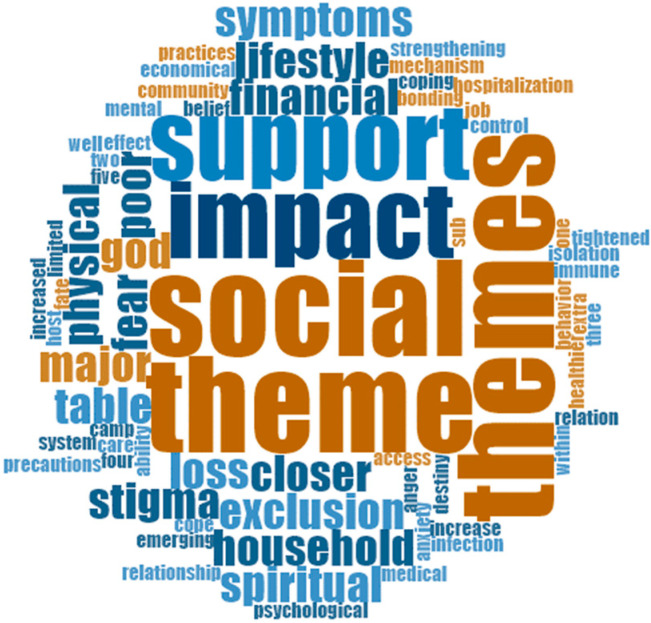
Word clouds.

**Table 1 ijerph-19-12588-t001:** Major themes reported from the qualitative assessment of the impact of COVID-19 on Syrian refugees.

Number	Major Themes	Sub-Themes
**Theme one**	Physical effect	**(a)** **Signs and Symptoms** **(b)** **Medical Support**
**Theme Two**	Mental and psychological impact	**(a)** **Stigma and Blame** **(b)** **Fear of Loss**
**Theme Three**	Social impact	**(a)** **Spouse support.** **(b)** **Seeking Support**
**Theme Four**	Spiritual impact	**(a)** **Closer to God** **(b)** **Acceptance**
**Theme five**	lifestyle and behavior	**(a)** **Precautions and extra protection** **(b)** **Vaccine hesitancy**

**Table 2 ijerph-19-12588-t002:** Emerging themes reported from the qualitative assessment of the impact of COVID-19 on Syrian refugees by setting.

Setting	
Camp	Host Community
Poor household support	Economical and financial impact
Social Exclusion	Fear of loss of financial support
Stigma	Limited access to medical care
Loss of Job	Closer and tightened social relationship within the same household
Poor social support	

**Table 3 ijerph-19-12588-t003:** Frequency of themes by setting (N: 20).

**Theme**	**Subtheme**	**Setting**	
		**Camp** **(N = 10)**	**Host Community** **(N = 10)**
Physical effect	Signs and Symptoms	8	8
	Good medical support	10	3
Mental and psychological impact	Stigma and Blame	5	4
	Fear of loss	4	6
Social impact	Bad spouse support	3	0
	Seeking support	5	6
Spiritual impact	Closer to God	10	10
	Acceptance	10	10
lifestyle and behavior	Precautions and extra protection	9	8
	Vaccine hesitancy	8	5

## Data Availability

Upon request to the corresponding author.

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
