# Peer review of "Exploring the Mental, Social, and Lifestyle Effects of a Positive COVID-19 Infection on Syrian Refugees in Jordan: A Qualitative Study"

_ijerph, 2022, doi:10.3390/ijerph191912588_

Round 1
Reviewer 1 Report
Dear Authors,
you touch a very important issue.
In your publication I miss a proof that your study describing twenty out of over million refugees can be generalized. It has a great risk of bias.
Your main conclusion "Living within the camp or host-community settings can be considered an essential determinant of how a positive COVID-19 diagnosis may impact the health and wellness of a Syrian refugee" is not supported by your study.
It lacks demographic and health information about refugees living in camp and non-camp settings in Jordan.
I agree that more studies on a larger scale would provide a better picture.
Author Response
| Yes | Can be improved | Must be improved | Not applicable | |
| Does the introduction provide sufficient background and include all relevant references? | ( ) | ( ) | (x) | ( ) |
| Are all the cited references relevant to the research? | (x) | ( ) | ( ) | ( ) |
| Is the research design appropriate? | (x) | ( ) | ( ) | ( ) |
| Are the methods adequately described? | (x) | ( ) | ( ) | ( ) |
| Are the results clearly presented? | ( ) | (x) | ( ) | ( ) |
| Are the conclusions supported by the results? | ( ) | ( ) | (x) | ( ) |
1.We would like to thank the reviewer for the comments. Please note that we have reviewed the Introduction and added more information about refugees in Jordan.
2. Our response to the comment, "In your publication I miss a proof that your study describing twenty out of over a million refugees can be generalized. It has a great risk of bias.":
Response:
In these types of papers, generalization is not sought. Usually, these are explorative studies that point out issues related to specific groups of people that may need further research. We never aimed at generalization with our study. As reported by Polit DF, Beck CT (2010)1. “The goal of most qualitative studies is not to generalize but rather to provide a rich, contextualized understanding of some aspect of human experience through the intensive study of particular cases”. Please check the below reference: Polit DF, Beck CT (2010). Generalization in quantitative and qualitative research: myths and strategies. Int J Nurs Stud. 2010 Nov;47(11):1451-8. doi: 10.1016/j.ijnurstu.2010.06.004. Epub 2010 Jul 3. PMID: 20598692.
2. Our response to the comment: "Your main conclusion "Living within the camp or host-community settings can be considered an essential determinant of how a positive COVID-19 diagnosis may impact the health and wellness of a Syrian refugee" is not supported by your study":
Response:
Thank you again for your comment. While we agree that we cannot exclusively indicate that this is the case, we have clearly stated that this can be one essential determinant of the impact of a positive diagnosis on this population.
Our study identified themes for a COVID-19 diagnosis among refugees within the camp and non-camp settings. It was reported that while some themes are shared, the impact of COVID-19 seems to differ by setting. This was clearly stated in more than one theme. As such, we think the results support the statement provided. We appreciate reconsidering the comment in line with this explanation. As a support, we here provide the text within the manuscript that supports our response to the comment above:
“Emerging themes
Those living within community settings had a financial and economic impact from COVID-19. Many specified that such financial impacts affected their access to testing, medical help, and coping ability. They experienced anger and fear of losing their job and the possibility of losing their income. Mothers or females in this group had a strong feeling of responsibility to look after their spouses, the breadwinners, and their children.
Those within camp settings experienced more social exclusion, and social issues with husbands and neighbours, affecting their mental health and social relationships. Those within camp settings did not report problems accessing medical care and medications. They had no fear of being jobless; only one or two mentioned this issue, and some said being affected financially.”
3. Our response to the comment: " It lacks demographic and health information about refugees living in the camp and non-camp settings in Jordan"
Response :
Thank you for your comment. We have added this in the introduction.

Reviewer 2 Report
A purposive and convenience sample of 20 refugees and five HCP (were this 3 MD and 2 Nurses) were provided. How were they selected? On what basis/criteria? Depending on a single privileged information to provide interviewees is not reassurant.
What about the ten refugees living in non- camp? How were they selected?
Sample selection needs more detailed explanaition.
Conclusions can (must) be more detailed otherwise the paper will be a sum of quotations on the interviews to refugees.
Author Response
1. Does the introduction provide sufficient background and include all relevant references?
(x) ( ) ( ) ( )
Are all the cited references relevant to the research?
(x) ( ) ( ) ( )
Is the research design appropriate?
(x) ( ) ( ) ( )
Are the methods adequately described?
(x) ( ) ( ) ( )
Are the results clearly presented?
( ) (x) ( ) ( )
Are the conclusions supported by the results?
( ) ( ) (x) ( )
Response:
Thank you for your comments. We have improved our results and conclusion presentation as suggested.
2. Our response to the comment: "A purposive and convenience sample of 20 refugees and five HCP (were this 3 MD and 2 Nurses) were provided. How were they selected? On what basis/criteria? Depending on a single privileged information to provide interviewees is not reassurant. What about the ten refugees living in non- camp? How were they selected? Sample selection needs more detailed explanation".
Response:
We thank the reviewer for this important comment. We have further explained the recruitment in the method section line.
The Researcher who conducted the interviews had access to the refugee camp through a doctor who worked there. The doctor invited refugees through his encounter with them to participate in the study. Those who agreed were referred to the researcher who contacted them, got their consent to participate and then interviewed them. As for those in the community (non-camp): they were recruited through the snowball technique i.e another existing study subjects recruit future subjects from among their acquaintances.
3.Our response to the comment: "Conclusions can (must) be more detailed otherwise the paper will be a sum of quotations on the interviews to refugees."
Response:
We thank the reviewer for the important comment on the conclusion. Please note that we have detailed our conclusion as suggested.

Reviewer 3 Report
This paper provides an early useful investigation of the effects of Covid for refugees. The general approach is particularly topical and useful because it addresses refugee experiences in a transitional country rather than a final resettlement country, and because inset out to compare experiences of refugees from one country who are transitioning and experienced (and recovered from) COVID in different circumstances (in the community or refugee camps). The sample is small, but in an initial study I think there is room for that. What I think the authors need to do is to make more of the community/camp distinctions of their participants, and I suggest a few things they could do to raise the level of their reporting. They make the availability of 2 subsamples part of their approach, and try to keep that alive with their identification system for individual participants.
I recommend that they add small quantitative element to their analyses. There is a subgroup comparison built into their approach but not followed as directly as possible in their analyses and reporting. It is however picked up in the Discussion Section - e.g., at line 469 - that statement would have a basis in the Results. Similarly throughout the Results, the authors refer to "many" and "most". There is no way the reader can connect to that or to check it.
So I recommend strongly that: 1. Calculate the numbers of each subgroup expressing each of the themes; and 2. Present these numbers in a simple table and follow the logic of the group differences as part of their qualitative reporting of the results.
I do not think this will compromise the qualitative analyses and reporting - in fact it will strengthen it. It is an acceptable way of presenting numbers for small samples. I do not think it needs quantitative analysis - there are equally 10 in the subsamples - you could present the numbers as percentages. This will give the authors a firmer basis for discussing the differences and make their assumed comparison explicit. The Results section would be more organised and the plausibility and scrutiny of the Discussion would be clearer. I would like to see this paper published. Yes it is a small scale study and they are asking to publish it in a high level international journal. I would advise building up the Limitations Section and pointing to the originality of the research and its initial value. A small additional table would assist this. I think it will make the qualitative analyses stronger and the results more accessible to a wider audience and add to the value of an otherwise well done qualitative study.
Author Response
- Reviewer’s Comment”
( ) I would not like to sign my review report
(x) I would like to sign my review report
English language and style
( ) Extensive editing of English language and style required
( ) Moderate English changes required
(x) English language and style are fine/minor spell check required
( ) I don't feel qualified to judge about the English language and style
Yes Can be improved Must be improved Not applicable
Does the introduction provide sufficient background and include all relevant references?
(x) ( ) ( ) ( )
Are all the cited references relevant to the research?
(x) ( ) ( ) ( )
Is the research design appropriate?
( ) (x) ( ) ( )
Are the methods adequately described?
( ) (x) ( ) ( )
Are the results clearly presented?
( ) (x) ( ) ( )
Are the conclusions supported by the results?
( ) (x) ( ) ( )
We would like to thank the reveiwer for the valuable comments .
2. Our response to the comment: ”This paper provides an early useful investigation of the effects of Covid for refugees. The general approach is particularly topical and useful because it addresses refugee experiences in a transitional country rather than a final resettlement country, and because inset out to compare experiences of refugees from one country who are transitioning and experienced (and recovered from) COVID in different circumstances (in the community or refugee camps). The sample is small, but in an initial study I think there is room for that. What I think the authors need to do is to make more of the community/camp distinctions of their participants, and I suggest a few things they could do to raise the level of their reporting. They make the availability of 2 subsamples part of their approach, and try to keep that alive with their identification system for individual participants.
I recommend that they add small quantitative element to their analyses. There is a subgroup comparison built into their approach but not followed as directly as possible in their analyses and reporting. It is however picked up in the Discussion Section - e.g., at line 469 - that statement would have a basis in the Results. Similarly throughout the Results, the authors refer to "many" and "most". There is no way the reader can connect to that or to check it.
So I recommend strongly that: 1. Calculate the numbers of each subgroup expressing each of the themes; and 2. Present these numbers in a simple table and follow the logic of the group differences as part of their qualitative reporting of the results.
I do not think this will compromise the qualitative analyses and reporting - in fact it will strengthen it. It is an acceptable way of presenting numbers for small samples. I do not think it needs quantitative analysis - there are equally 10 in the subsamples - you could present the numbers as percentages. This will give the authors a firmer basis for discussing the differences and make their assumed comparison explicit. The Results section would be more organised and the plausibility and scrutiny of the Discussion would be clearer. I would like to see this paper published. Yes it is a small scale study and they are asking to publish it in a high level international journal. I would advise building up the Limitations Section and pointing to the originality of the research and its initial value. A small additional table would assist this. I think it will make the qualitative analyses stronger and the results more accessible to a wider audience and add to the value of an otherwise well-done qualitative study."
Response:
We Thank the reviewer for pointing out this valuable addition to the paper. We have included a table (Table 3.) that provides quantitative distribution of themes among participants by numbers and percentages. We have also strengthened the results and discussion sections, as suggested.

Reviewer 4 Report
The article discusses the Covid-19 pandemic impact on the mental health of vulnerable subjects. Such thematic is of interest, since not much has been written regarding the effect of the pandemic on refugee population.
Furthermore, this subject is more interesting considering the special risk that this population suffers due to their particular life condition.
However, the article could be improved with some revisions, taking into consideration the following points:
The division of the results in “themes” is appropriate and very interesting, also capturing the reader attention by citing the refugees’ interviews. However, the comments of the researchers is scarce and it looks like a list of interview extracts. I do believe that more interpretation and comments must be added by the authors in order to enrich the article.
The article talks about the vaccine hesitancy, however no data is given about the vaccination of the interviewees. I guess that at this stage of the pandemic such data should be provided, also to better understand the part of the article that regards such hesitancy.
I do appreciate that the authors compiled a list of limitations of the study, which I suggest should include more clearly the reason of the little number of interviews, in order to justify that.
Finally, I find that the bibliography is somehow poor and it could definitely be enriched, as there are many relevant works that can enrich and strengthen the state of art of the article.
For example, the mental health of refugees in Jordan has been studied in relation to the difficulty of integration and education in a work titled “Refugees education: an ethnography of teaching experiences in Jordan”.
Author Response
Reviewer 4
- Reviewer’s Comment
Yes Can be improved Must be improved Not applicable
Does the introduction provide sufficient background and include all relevant references?
( ) (x) ( ) ( )
Are all the cited references relevant to the research?
( ) ( ) (x) ( )
Is the research design appropriate?
( ) (x) ( ) ( )
Are the methods adequately described?
(x) ( ) ( ) ( )
Are the results clearly presented?
( ) (x) ( ) ( )
Are the conclusions supported by the results?
( ) (x) ( ) ( )
Authors’ Response:
The article discusses the Covid-19 pandemic impact on the mental health of vulnerable subjects. Such thematic is of interest, since not much has been written regarding the effect of the pandemic on refugee population.
Furthermore, this subject is more interesting considering the special risk that this population suffers due to their particular life condition.
Response: Thank you very much for your valuable comments . We agree that not much has been done in this area . We thank you also for the feedback that will definitely help in improving the quality of our paper.
- Reviewer’s Comment
However, the article could be improved with some revisions, taking into consideration the following points:
The division of the results in “themes” is appropriate and very interesting, also capturing the reader attention by citing the refugees’ interviews. However, the comments of the researchers is scarce and it looks like a list of interview extracts. I do believe that more interpretation and comments must be added by the authors in order to enrich the article.
Authors’ comments
We have included our own interpretations to the themes as recommended. Please see results section
- Reviewer’s Comment
The article talks about the vaccine hesitancy, however no data is given about the vaccination of the interviewees. I guess that at this stage of the pandemic such data should be provided, also to better understand the part of the article that regards such hesitancy.
Authors’ comments
We agree with you, but unfortunately, vaccination was not part of the scope of this paper. The time of the data collection, vaccinations efforts were not intensified. We agree that further studies on refugees vaccination are needed. We have added this to our recommendations. We have also updated the text with vaccination percentages among camp and non-camp refugees. This meets some of the concerns raised by the reviewer.
- Reviewer’s Comment
I do appreciate that the authors compiled a list of limitations of the study, which I suggest should include more clearly the reason of the little number of interviews, in order to justify that.
Authors’ comments
Thank you for the reviewer’s comments. We have added to the limitations an explanation on why the number is small. During COVID-19, online research was the only option as social encounters were almost restricted and research was not allowed to be face to face.
- Reviewer’s Comment
Finally, I find that the bibliography is somehow poor and it could definitely be enriched, as there are many relevant works that can enrich and strengthen the state of art of the article.
For example, the mental health of refugees in Jordan has been studied in relation to the difficulty of integration and education in a work titled “Refugees education: an ethnography of teaching experiences in Jordan”.
Authors’ comments
Thank you for pointing out this important point. We have updated our references and added a couple of new references to both the introduction and discussion paragraphs.
Thank you again

Round 2
Reviewer 3 Report
The authors have refined their MS and it is nearly ready for publication. However, I think there is a little more they could do to make the findings clearer and more relevant. Specifically I recommend
(1) refine Table 3. I see that Table 2 is a summary Table, but I think it needs reading into the Results. You only need one column for each setting. Since both are N=10, I would put N=10 in Header for each and then out numbers in the cells – e.g., 8, 8.
(2) insert a comment about Tables 2 and 3 at l. 188, and
(3) Refer to Table 3, reading off Tables 2 and 3 in overview the similarities in the 2 settings, and the particular difference of vaccine hesitancy (8 in Camp and 5 in Host Community), and also Social impact (3 in Camp and none in Host Community). I suggest this should go at about l. 204 as a short overview. I am not sure you need Table 2, it could be part of the text as an overview.
(4) Then could you pick up these differences in the Discussion, probably around l. 532. The vaccine hesitancy in the camp is interesting, especially if the authors have some suggestion to why this is so.
I think these extra refinements enhance the accessibility of the information and the usefulness of the paper.
Author Response
The authors have refined their MS and it is nearly ready for publication. However, I think there is a little more they could do to make the findings clearer and more relevant. Specifically I recommend:
1. refine Table 3. I see that Table 2 is a summary Table, but I think it needs reading into the Results. You only need one column for each setting. Since both are N=10, I would put N=10 in the Header for each and then out numbers in the cells – e.g., 8, 8.
Authors’ Comment
We thank the reviewer, we have refined table 3 as per the reviewer’s suggestion
2. Insert a comment about Tables 2 and 3 at l. 188, and
Authors’ Comment
We thank the reviewer. Please note that, as highlighted, we have inserted comments about the two tables as per the reviewer’s suggestions in the result section (currently lines 206-208).
3.Refer to Table 3, reading off Tables 2 and 3 in overview the similarities in the 2 settings, and the particular difference of vaccine hesitancy (8 in Camp and 5 in Host Community), and also Social impact (3 in Camp and none in Host Community). I suggest this should go at about l. 204 as a short overview. I am not sure you need Table 2, it could be part of the text as an overview.
Authors’ Comment
We thank the reviewer. Please note that we have inserted comments about the two tables as per the reviewer’s suggestions in the result section (currently line 204-206) as highlighted before. We would like to reiterate that table2.is presenting emerging themes that were not part of the original research question, and we think it would be helpful to keep it as a table in the manuscript. We agree with the reviewer that it could be highlighted in the text, and we have done that in section 3.B Line 446-464.
4. then could you pick up these differences in the Discussion, probably around l. 532. The vaccine hesitancy in the camp is interesting, especially if the authors have some suggestions as to why this is so.
I think these extra refinements enhance the accessibility of the information and the usefulness of the paper.
Authors’ Comment
We thank the reviewer for these important suggestions. Please note that we have discussed the vaccine hesitancy and social support in discussion lines-552-574.
Thank you, and we highly appreciate your thorough review.
